# A Novel Mutation Related to Aceruloplasminemia with Mild Clinical Findings: A Case Report

**DOI:** 10.3390/reports8010004

**Published:** 2024-12-31

**Authors:** Alexandros Giannakis, Tsamis Konstantinos, Maria Argyropoulou, Georgia Xiromerisiou, Spiridon Konitsiotis

**Affiliations:** 1Department of Neurology, Faculty of Medicine, School of Health Sciences, University of Ioannina, 45500 Ioannina, Greece; papadates@gmail.com (A.G.); skonitso@uoi.gr (S.K.); 2Department of Physiology, Faculty of Medicine, School of Health Sciences, University of Ioannina, 45500 Ioannina, Greece; 3Department of Clinical Radiology, Faculty of Medicine, School of Health Sciences, University of Ioannina, 45500 Ioannina, Greece; margyrop@uoi.gr; 4Department of Neurology, Faculty of Medicine, University Hospital of Thessaly, 45100 Larissa, Greece; georgiaxiromerisiou@gmail.com

**Keywords:** aceruloplasminemia, neurodegeneration with brain iron accumulation, ceruloplasmin, ferritin, case report

## Abstract

**Background and Clinical Significance**: Aceruloplasminemia (ACP), a member of the neurodegeneration with brain iron accumulation (NBIA) spectrum of disorders, is a rare disorder caused by mutations in the ceruloplasmin (CP) gene. Iron accumulation in various organs, including the brain, liver, eyes, and heart, can lead to a broad clinical spectrum. Here, we report the first case of ACP in Greece. **Case Presentation**: Our patient was a 53-year-old male who was referred to our movement disorders center for a 6-month history of mild, unspecific, episodic dizziness and postural instability, and attention and memory deficits. Brain MRI revealed significant iron accumulation in multiple brain regions, including the dentate nuclei, cerebellar cortex, basal ganglia, thalamus, brainstem nuclei, and hypothalamus. These findings were particularly evident in susceptibility-weighted images. Fundoscopy revealed a normal retina, optic nerve, and macula. Whole-exome sequencing revealed a novel homozygous frameshift mutation in the CP gene [NM_000096.3:p.Thr3232fs (c.9695delC)]. This mutation has not been previously reported and is predicted to result in premature protein termination, supporting its pathogenic nature. Laboratory tests showed no anemia but revealed significantly elevated serum ferritin and low serum iron. Subsequent testing revealed extremely low serum CP and low serum copper. Despite less involvement of the myocardium, our patient succumbed to cardiac arrest. **Conclusions**: ACP should be considered in cases with minor neurological signs and symptoms. Brain MRI plays a significant role in early diagnosis. Close cardiac monitoring is also important.

## 1. Introduction and Clinical Significance

Aceruloplasminemia (ACP) is a rare, autosomal recessive disorder caused by mutations in the ceruloplasmin (CP) gene [1]. CP, a copper-containing ferroxidase enzyme, is crucial for extracellular iron transport and may have neuroprotective properties [1]. ACP is a member of the neurodegeneration with brain iron accumulation (NBIA) spectrum of disorders [2]. Typically, neurological manifestations, such as cerebellar symptoms, extrapyramidal signs, involuntary movements, and cognitive–psychiatric disorders, begin between the fifth and sixth decade of life [1,2]. Iron accumulation in various organs, including the brain, liver, eyes, and heart, can lead to a broad clinical spectrum, including microcytic anemia, diabetes mellitus, peripheral retinal degeneration, and cardiac disease [1,2]. However, despite extensive iron overload, ACP can present with subtle clinical findings [3]. Here, we report the first case of ACP in Greece, a patient with a novel mutation in the CP gene who presented with mild symptoms but significant iron accumulation in multiple organs.

## 2. Case Presentation

Our patient was a 53-year-old male who was referred to our movement disorders center for a 6-month history of mild, unspecific, episodic dizziness and postural instability. He also complained of attention and memory deficits. His past medical and family history was unremarkable.

Clinical examination revealed only a mild, bilateral action tremor in his upper limbs, which had both postural and kinetic features. Neurocognitive assessment showed minor deficits in executive function, attention, and visuospatial abilities (Addenbrooke’s Cognitive Examination—Revised 97/100, Frontal Assessment Battery 17/18).

To our surprise, brain MRI (Figure 1) revealed significant iron accumulation in multiple brain regions, including the dentate nuclei, cerebellar cortex, basal ganglia, thalamus, brainstem nuclei, and hypothalamus. These findings were particularly evident in susceptibility-weighted images (SWI). The cerebral cortex also showed evidence of iron accumulation, albeit to a lesser extent. Additionally, the MRI demonstrated cerebral and cerebellar cortical atrophy, disproportionate to the patient’s age. Pituitary gland evaluation revealed hypopituitarism, with a gland size of 3.2 mm.

**Figure 1 reports-08-00004-f001:**
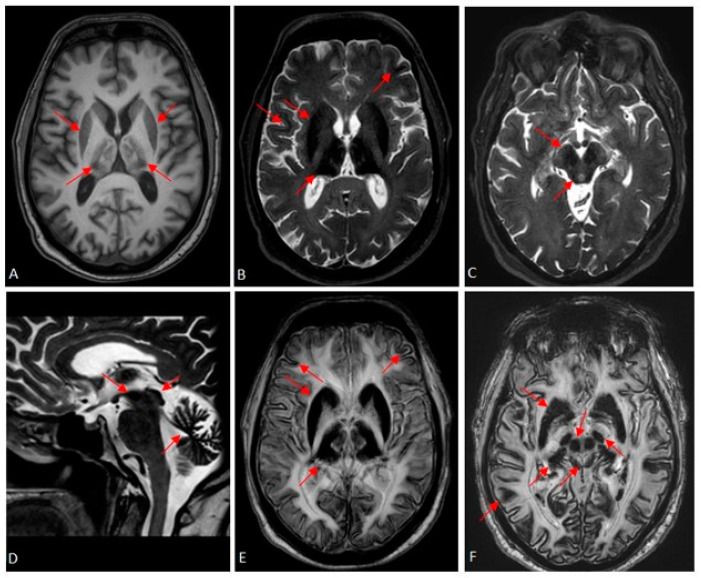
Brain MRI T1-weighted (**A**), T2-weighted (**B**–**D**), and susceptibility-weighted (**E**,**F**) images that demonstrate excessive iron deposition at the caudate nuclei, putamen, thalami, substantia nigra, red nuclei, midbrain tectum, dentate nuclei, and cerebellar cortex (red arrows).

Electroencephalography, nerve conduction studies, and electromyography were all unremarkable. Ophthalmological evaluation with fundoscopy and electroretinography was performed, revealing a normal retina, optic nerve, and macula.

Table 1 summarizes the main laboratory findings. Laboratory tests showed no anemia but revealed significantly elevated serum ferritin and low serum iron. Subsequent testing revealed extremely low serum CP and low serum copper.

Hormonological assessment revealed normal thyroid function and no evidence of pituitary insufficiency. Glycosylated hemoglobin was elevated at 6.2 mg/dL. C-peptide levels were within the normal range (1.6 mg/dL), and insulin-like growth factor 1 levels were low, suggesting a combination of insulin secretion disorder and insulin resistance.

Abdominal computed tomography (CT) revealed pancreatic lipomatosis and multiple punctate calcifications. Additionally, the liver was significantly reduced in size with increased echogenicity, despite normal liver function tests. A liver biopsy was performed to further investigate these findings.

Despite a normal electrocardiogram and transthoracic echocardiogram, we proceeded to cardiovascular MRI (CMR) due to a high suspicion of ACP and extensive iron accumulation in other organs, despite mild clinical findings and limited laboratory abnormalities. CMR demonstrated mild midventricular hypokinesis and cardiac dilatation of the pulmonary outflow tract. Fast field echo MRI demonstrated significant iron accumulation, primarily in the basal ganglia and brainstem, with lesser involvement of the myocardium and liver.

Given the high clinical suspicion of ACP, we performed whole-exome sequencing (WES) using ION Torrent^TM^ (Thermo Fisher Scientific Inc., Waltham, US) semiconductor sequencing and analyzed the results with ClinGenics (London, UK) Exome Management Application^®^ Disease Relevance (EMADR). The analysis revealed a novel homozygous frameshift mutation in the CP gene [NM_000096.3:p.Thr3232fs (c.9695delC), Online Mendelian Inheritance in Man (OMIM^®^) *117700, cytogenetic location: 3q24-q25.1]. This mutation has not been previously reported in any international genetic database and is predicted to result in premature protein termination, supporting its pathogenic nature.

Following the confirmation of the ACP diagnosis through genetic analysis, we recommended genetic testing for the patient’s relatives, but they declined due to financial constraints. Expanding government subsidies for genetic testing can make it more affordable and accessible to a wider population.

Our follow-up plan was to start the patient on iv deferiprone and included monthly liver function tests, ferritin, and C-reactive protein, quarterly glycosylated hemoglobin, and annual ophthalmological evaluations. We also planned a six-month reevaluation with a neurological examination and brain MRI. Tragically, our patient suffered two cardiac arrests and died while en route to our hospital for revaluation. Despite less severe cardiac iron accumulation compared to the brain and liver, the patient’s heart ultimately failed.

## 3. Discussion

ACP is an extremely rare disease. Its prevalence is estimated at 1:2,000,000 [1,4]. While most of the roughly 100 reported cases originate from Japan [4], cases have also been documented in China and various European countries, including Italy, Spain, France, Germany, Denmark, Belgium, and Poland [1,3,4,5]. This marks the first known case of ACP in Greece.

CP is a copper-binding glycoprotein that oxidizes Fe²⁺ to Fe³⁺, facilitating iron mobilization from cells and the subsequent binding to transferrin [6]. During hepatic synthesis, six copper atoms bind to CP, essential for both its stability and function, as a copper-deficient CP is rapidly degraded after it enters plasma [4]. CP exists in two isoforms. The soluble isoform, primarily synthesized by hepatocytes, is involved in nitric oxide homeostasis [1]. The second isoform is a glycosylphosphatidylinositol (GPI)-anchored membrane protein found on the surface of various cells, including hepatocytes as well as pancreatic and retinal epithelial cells [1]. In the brain, it is predominantly expressed in astrocytes, ependymal cells, and pia mater cells, particularly in regions with high intracellular iron accumulation, such as the basal ganglia, thalamus, and dentate nucleus [6]. This membrane-bound form plays a crucial role in cellular iron mobilization: not only does it cooperate with transmembrane ferroportin to transport Fe^2+^ outside of the cells, but it also affects the modulatory role of hepcidin to ferroportin [1]. Furthermore, CP is an acute-phase reactant. Beyond its ferroxidase activity, which prevents the formation of reactive oxygen species, CP exhibits several antioxidant properties, including the inhibition of xanthine oxidase and lipid peroxidation, as well as the induction of nitric oxide synthase [7].

A gene mutation in the CP gene, resulting in decreased or absent enzymatic activity and CP levels, leads to impaired iron efflux, reduced circulating iron, deficient erythropoiesis, intracellular iron accumulation, and subsequent cellular damage [4]. The main organ-targets are the brain, liver, pancreas, heart, and retina [1,4]. Typically, systemic findings precede neurologic findings by approximately ten years and include mild microcytic anemia, low serum iron, high serum ferritin, low transferrin saturation, diabetes mellitus, and degenerative changes in the retina [4]. Neurological signs and symptoms typically manifest between the ages of 40 and 60 and usually progress to death [1,4]. These include movement disorders such as cerebellar ataxia, dysarthria, parkinsonism, oromandibular dystonia, chorea, and blepharospasm [1,7,8]. Cognitive, behavioral, and mood disorders are also common, especially in Caucasian patients [1,4].

The diagnosis is challenging. The classic triad of dementia, retinal degeneration, and diabetes is indicative of ACP [8]. However, recent European case series have reported a wider range of clinical and/or biochemical manifestations [4,5,9]. Laboratory tests can be helpful. Usually, the first biochemical sign of ACP is mild microcytic anemia. The combination of hypochromic microcytic anemia and atypically high serum ferritin should raise suspicion of deficient copper metabolism [10]. Additionally, the biochemical triad of microcytic anemia, elevated serum ferritin, and decreased transferrin saturation is suggestive of ACP, particularly after ruling out more common conditions like thalassemia and anemia of inflammation [1]. Other rare diseases, including divalent metal transporter 1 deficiency, atranferrinemia, congenital sideroblastic anemias, and ferroportin disease should also be excluded [11]. Low serum hepcidin can further differentiate ACP from other iron-overload disorders [5].

Low serum CP is also seen in Wilson disease, Menkes disease, hypoproteinemia, and even asymptomatic ACP heterozygotes. However, these have different clinical features from ACP, and, in contrast to them, CP is often characteristically low or absent in ACP [3,4,5,8,12,13]. However, in extremely rare cases, CP may be marginally low or even normal [1].

Diabetes is a common complication in individuals with ACP, often necessitating insulin therapy [1,9,10]. While typically manifesting in the fifth decade of life, juvenile-onset cases have been reported. Therefore, clinicians should consider ACP in young patients with type 1 diabetes and unexplained microcytic anemia [10]. Central diabetes insipidus and hypothalamic hypothyroidism have also been described in ACP [14].

Brain MRI reveals low signal in T2-weighted and, more prominently, susceptibility-weighted imaging (SWI) sequences within the caudate nuclei, putamen, thalamus, substantia nigra, red nuclei, midbrain tectum, dentate nuclei, and cerebellar cortex [13]. Other findings, such as putaminal cavitation, have also been described [15].

Treatment options for ACP are limited and controversial. Iron chelation therapy, using deferoxamine, deferasirox, and/or deferiprone is the mainstay of treatment [1]. Notably, deferiprone is the only iron chelator capable of crossing the blood–brain barrier (BBB) [1,16]. Some clinicians have explored additional therapeutic approaches, such as combining iron chelation with fresh frozen plasma [17] or phlebotomy [18]. While these strategies have shown some promise in certain cases [16,17,18,19], their efficacy remains uncertain [1]. Nevertheless, given their potential neuroprotective effects, they may be considered to mitigate neurological symptoms and progression [1]. Tetracyclines, such as minocycline, have been explored due to their iron-chelating properties and ability to cross the blood–brain barrier [20]. Additionally, some improvement has been observed with antioxidant therapies, including vitamin E and zinc sulfate [4,13].

Our patient was presented with a clinical picture that diverged from the classic ACP phenotype. Notably, he exhibited only mild neurological symptoms, lacked signs of retinal degeneration, and had prediabetes type 2 rather than overt diabetes. Consequently, he did not manifest the full spectrum of the classic ACP clinical triad [1]. Minor neurological findings have been previously described in ACP [3]. Interestingly, neurological involvement can be entirely absent in some cases [21]. Moreover, macular degeneration may not be present in all patients with ACP [22,23]. Given that low or absent serum CP levels are observed in 0.2% of the general population [3], it is plausible that the true prevalence of ACP may be higher than currently estimated [8]. Additionally, it suggests that subclinical or mild cases of the disorder may be more common than previously recognized.

Additionally, the patient did not exhibit anemia. Consequently, during his initial evaluation, ferritin, transferrin, CP, and glycosylated hemoglobin levels were not assessed. The only significant finding indicative of an NBIA disorder was the unexpected iron deposition in the basal ganglia and cerebellum, as revealed by brain MRI. This finding prompted further investigation and ultimately led to the genetic diagnosis of ACP. Therefore, we propose that any patient presenting with neurological symptoms or signs, even mild ones, should undergo a brain MRI to rule out potentially devastating conditions such as ACP. Early diagnosis and initiation of treatment are crucial for optimal patient outcomes. Lastly, despite the relatively minor cardiac muscle involvement, the patient tragically succumbed to cardiac arrest. Concurrent treatment with deferoxamine and deferiprone has shown some improvement in cardiac iron-loading [24]. Therefore, the extensive evaluation and close monitoring of cardiac function are of utmost importance in patients with ACP, even in the absence of overt signs of cardiac disease or cardiac iron overload.

## 4. Conclusions

In conclusion, the identification of a novel frameshift mutation in the first known Greek patient with ACP highlights the importance of considering this disease even in cases with only minor neurological signs and symptoms, without the typical systemic manifestations such as anemia and retinal degeneration. Brain MRI plays a crucial role in revealing the characteristic iron accumulation in the basal ganglia and cerebellum, facilitating early diagnosis. Moreover, close cardiac monitoring is essential, even in patients without overt cardiac iron overload, to prevent potentially fatal cardiac events.

## Figures and Tables

**Table 1 reports-08-00004-t001:** Main laboratory findings.

Serum Test	Patient’s Values	Normal Range
Hemoglobin	13.8 g/dL	13.5–15.5 g/dL
MCV ^1^	85.8 fL	80–100 fL
MCH ^2^	27.2 pg/cell	27–33 pg/cell
MCHC ^3^	31.7 g/dL	32–36 g/dL
Ferritin	1573 ng/mL	13–70 ng/mL
Iron	43 μg/mL	55–150 μg/mL
Ceruloplasmin	3 mg/dL	>20 mg/dL
Copper	5.6 μg/dL	10–15 μg/dL
Glycosylated hemoglobin	6.2%	<5.7%
C-peptide	1.6 ng/mL	0.5–3.3 ng/mL
Insulin	9.7 μIU/mL	1.9–23.0 μIU/mL

^1^ MCV: mean corpuscular volume; ^2^ MCH: mean corpuscular hemoglobin; ^3^ MCHC: mean corpuscular hemoglobin concentration.

## Data Availability

The data that support the findings of this study are not openly available due to reasons of sensitivity and are available from the corresponding author upon reasonable request. Data are located in controlled access data storage at the University Hospital of Ioannina.

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
