# Peer review of "A Novel Mutation Related to Aceruloplasminemia with Mild Clinical Findings: A Case Report"

_reports, 2024, doi:10.3390/reports8010004_

Round 1
Reviewer 1 Report
Comments and Suggestions for Authors
This is an interesting report on the first case of ACP in Greece. A homozygous frameshift mutation was presented in the CP gene and significant iron accumulation was revealed in multiple regions. Although only a low level of iron accumulation was found in the patient's heart, the patient passed away from heart failure. Thus, the authors concluded that cardiac monitoring may be essential for those ACP patients who may not have classic clinical presentations. The article is well presented, and the work is sufficient thus I endorse publication of this manuscript upon fixing one small point which is Line 114, the prevalence should be 1 : 2,000,000.
Comments on the Quality of English Language
The quality of English is sufficient.
Reviewer 2 Report
Comments and Suggestions for Authors
The quality of English must be improved:
line 23/35: instability...? Postural?
line 27/75: macular examinations
Line 33-35: you can ameliorate the conclusions
Line 58: thorough
Line 94: dilation
Why didn't you decide to finance the genetic testing of brother/sister?
It is necessary ameliorate the discussion. In particular line 137-139:
are you sure that ACP is the only known disease in which iron accumulates in other organ systems?
Comments on the Quality of English Language
The quality of English must be improved. In particular: line 23/35, 27/75, 33-35, 58, 94.
Round 2
Reviewer 2 Report
Comments and Suggestions for Authors
Could you add *117700 (OMIM classification) and the cytogenetic location (3q24q 25.1) of CP gene?
Author Response
Dear MDPI author services,
Sincerely thank you for giving us the opportunity to submit a revised draft of our manuscript (ID: reports-3323122) entitled "A Novel Mutation Related to Aceruloplasminemia with Mild Clinical Findings: A Case Report" to Reports. We again appreciate the time and effort that you and the reviewers have dedicated for a second time to providing your valuable feedback on our manuscript. We are grateful to the reviewers for their insightful comments on our paper. We have been able to incorporate changes to reflect the suggestions provided by the reviewers. As suggested, all changes are highlighted in yellow in the revised version of our manuscript. Here is a point-by-point response to the reviewers’ comments and concerns.
Reviewer #2:
Comment 1: “Could you add *117700 (OMIM classification) and the cytogenetic location (3q24q 25.1) of CP gene?”
Thank you for your suggestion. Accordingly, we have added OMIM classification and cytogenetic location of ceruloplasmin gene (lines 102-103).
We look forward to hearing from you in due time regarding our submission and to responding to any further questions and comments you may have.
Yours sincerely,
Alexandros Giannakis, MD, MSc, PhD
Attending Neurologist
Neurology Department
University Hospital of Ioannina